# PROMPT2BOX: Uncovering Entailment Structure among LLM Prompts

## Abstract

To discover the weaknesses of LLMs, researchers often embed prompts into a vector space and cluster them to extract insightful patterns. However, vector embeddings primarily capture topical similarity. As a result, prompts that share a topic but differ in specificity, and consequently in difficulty, are often represented similarly, making fine-grained weakness analysis difficult. To address this limitation, we propose PROMPT2BOX, which embeds prompts into a box embedding space using a trained encoder. The encoder, trained on existing and synthesized datasets, outputs box embeddings that capture not only semantic similarity but also specificity relations between prompts (e.g., "*writing an adventure story*" is more specific than "*writing a story*"). We further develop a novel dimension reduction technique for box embeddings to facilitate dataset visualization and comparison. Our experiments demonstrate that box embeddings consistently capture prompt specificity better than vector baselines. On the downstream task of creating hierarchical clustering trees for 17 LLMs from the UltraFeedback dataset, PROMPT2BOX can identify 8.9% more LLM weaknesses than vector baselines and achieves an approximately 33% stronger correlation between hierarchical depth and instruction specificity.

## 1. Introduction

When developing a large language model (LLM), it is crucial to identify its weaknesses to guide the collection of additional high-quality training data. Several prior works (Jiang et al., 2024; Tamkin et al., 2024; Zeng et al., 2025; Tian et al., 2025) address this practical requirement by first embedding every prompt into a vector and hierarchically cluster the

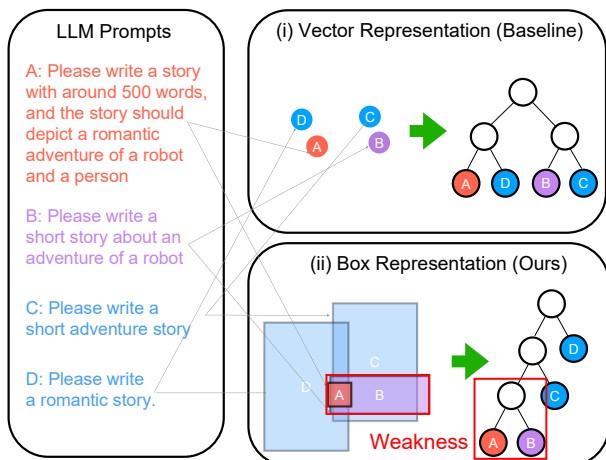

*Figure 1.* Comparison between the widely-used vector representation and our box representation for analyzing the performance of an LLM on four prompts. Blue means that the LLM achieves a high performance on the prompt while red means the opposite. Our approach correctly highlights that a weakness of LLM is *writing an robot adventure* by clustering prompt *A* and *B*.

prompts into a tree based on their similarities. Then, by analyzing performance differences across these clusters or across regions in a two-dimensional projection of the embedding space, LLM developers can diagnose systematic weaknesses relative to competing models.

The vector-based methods rely on the assumption that LLMs perform similarly for similar prompts, without accounting for the degree of difficulty involved with the prompt. Recent studies (Atmakuru et al., 2024; Lu et al., 2025; Jaroslawicz et al.; Zhang et al., 2025b) show that adding constraints to a prompt reduces the solution space, making the prompt more specific and more difficult. However, the current vector-based approaches cannot model this important specificity information of the prompt, so they often group two prompts that are topically similar but have different difficulties. Ignoring the prompt specificity brings undesired ambiguities for interpreting the LLMs' performance using the vector-based approach. For example, when a developer observes a low average score for a prompt cluster, it is unclear whether the LLM performs poorly on the underlying topic in gen-

[1]Anonymous Institution, Anonymous City, Anonymous Region, Anonymous Country. Correspondence to: Anonymous Author <anon.email@domain.com>.

Preliminary work. Under review by the International Conference on Machine Learning (ICML). Do not distribute.

eral, or whether it struggles only with the more specific or difficult prompts within that cluster.

Figure 1 (i) illustrates an example. $A$ is semantically close to $D$, but being much more specific than $D$. The low score of $A$ decreases the average score of the cluster that corresponds to $A$ and $D$, so the cluster becomes a weakness of this LLM. This cluster should represent LLMs' ability of "*writing a romantic story*", but $D$ has shown that the LLM could write a pretty good *romantic story*. This example highlights the limitation of only considering the similarity in the LLMs' weakness analysis task.

To address this problem, we propose PROMPT2BOX, which embeds each prompt into a high-dimensional box embedding space. Unlike vector embeddings, box embeddings (Vilnis et al., 2018) can naturally represent asymmetric semantic relationships, such as entailment, making them well-suited for modeling hierarchical structure among prompts. Conceptually, each textual prompt is represented by a box parameterized by a center vector and a size vector. The center vector captures the semantic location of the prompt, such that semantically similar prompts are mapped to nearby centers. The size vector controls the semantic scope of the prompt: more general prompts are represented by larger boxes, while more specific prompts correspond to smaller boxes. This geometry allows entailment relationships to be expressed through box containment.

Specifically, if the box corresponding to one prompt is contained within the box of another prompt, we interpret this as an entailment relation, where the more specific prompt entails the more general one. For example, in Figure 1(ii), box $A$ is almost entirely contained within box $D$, reflecting that the prompt "*writing a romantic adventure story*" (prompt $A$) semantically entails the more general prompt "*writing a romantic story*" (prompt $D$).

We show that the box of a prompt could also be interpreted as the space of its valid responses. In this interpretation, a good performance of an LLM for a prompt means that there exists a good response in this box that could be outputted by this LLM, but the LLM is still likely to output some other bad responses in this box. For example, the LLM does badly for prompt $A$ and $B$ but not for prompt $D$, which suggests that the LLM is not good at "*writing an adventure story of a robot*", especially when the adventure involves some *romantic* relations, but this LLM is probably good at writing other kinds of *romantic stories*. Our box representation demonstrates that the weakness of the LLM lies on the region of box $A$ and $B$, while we cannot get the similar conclusion from the vector baseline because of its lack of specificity information.

To discover the entailment structure among prompts, we leverage the existing entailment datasets and synthesize more entailment relationships between prompts. Next, we train an encoder to map every prompt into a box. Furthermore, we propose a new dimension reduction method and a new hierarchical clustering method for box. Our experiments show that our box embeddings predict the entailment relationships much better than the vector baselines, which allows us to better analyze the weaknesses of LLMs through a 2D box embedding space and our specificity-aware hierarchical clustering method.

### 1.1. Main Contributions

- We propose PROMPT2BOX, which uses a box embedding-based representation to capture the entailment relation among prompts.
- We propose novel methods to synthesize entailment datasets for training an encoder that converts each LLM prompt into a box embedding.
- We propose BOX-SNE, a novel multiple dimension compression method for box embeddings, and a new hierarchical clustering algorithm for box embeddings.
- We propose new evaluation metrics to assess similarity, entailment, and specificity in prompt representations. We further demonstrate how box embeddings can be used to analyze LLMs as well as LLM evaluation benchmarks.

## 2. Related Work

Box embeddings (Vilnis et al., 2018), a form of region-based embeddings, have been shown to outperform other region-based representations such as Order Embeddings (Vendrov et al., 2016) and Poincaré Embeddings (Nickel & Kiela, 2017) in modeling asymmetric relationships. Box embeddings have been successfully applied to model hierarchical and structured semantic relationships across multiple domains. In computer vision, Daroya et al. (2024) use box embeddings to represent task-level hierarchies. In the context of knowledge bases, box embeddings effectively capture hierarchical graph structures such as WordNet (Patel et al., 2020; Dasgupta et al., 2021) and OWL ontologies (Jackermeier et al., 2024). Furthermore, Ren et al. (2020); Dasgupta et al. (2021) introduce box-embedding-based formulations for knowledge graph query answering, where the logical structure of a query is directly encoded in the embedding space. As far as we know, no work uses box to analyze prompts or LLMs' weaknesses.

As identifying LLMs' weaknesses becomes increasingly important, more and more benchmark/prompt analysis methods are proposed. Examples include Clio (Tamkin et al., 2024), SkillVerse (Tian et al., 2025), and EvalTree (Zeng et al., 2025). Moreover, many recent studies leverage LLMs to discover taxonomy and categories from a corpus (Hsu

et al., 2024; Tian et al., 2024; Zhang et al., 2025a; Kargupta et al., 2025; Zhong et al., 2025; Gao et al., 2025; Chirkova et al., 2025). Although different papers might use different clustering methods or leverage LLMs in different ways, most of them conduct (hierarchical) clustering based on a vector embedding space. Our paper discovers that boxes could perform better than vectors in terms of identifying LLMs' weakness clusters, and thus could potentially improve over the aforementioned related works.

## 3. Method

We first introduce the definition of entailment between prompts and establish the connection among constraint space, entailment, and solution space in Section 3.1. Next, we introduce our ways of parameterizing the box embedding and computing the intersection size and entailment probability in Section 3.2. In Section 3.3, the details of optimizing our encoder are described. Finally, Section 3.4 explains how the training data are curated.

### 3.1. Definition of Terms

Let $\mathcal{X}$ denote the space of instructions. Let $\mathcal{U}$ denote the universe of all possible constraints and we assume the number of possible constraints $|\mathcal{U}|$ is finite. For any instruction $x \in \mathcal{X}$, let $\mathcal{C} : \mathcal{X} \to 2^{\mathcal{U}}$ be a mapping from an instruction to the set of constraints it induces, where

$$\mathcal{C}(x) = \{\, c \mid c \in \mathcal{U} \text{ is satisfied by all valid responses to } x \,\}. \quad (1)$$

Recall that for any two statements $g$ and $h$, if $g$ entails $h$ (written $g \models h$), then whenever $g$ is true, $h$ must also be true. In other words, $g$ imposes a stronger condition than $h$. When $g$ and $h$ are both prompts, $g \models h$ means asking prompts $g$ implies asking prompts $h$. Formally, constraint inclusion is the same as entailment between the two prompts:

$$\forall\, a, b \in \mathcal{X}, \qquad \mathcal{C}(a) \supseteq \mathcal{C}(b) \iff a \models b. \quad (2)$$

Furthermore, we say the prompt $a$ is more specific than prompt $b$ if the prompt $a$ has more constraints (i.e., $C(a) \supset C(b)$). Taking the example present in Figure 1, we have two prompts: $g =$ "*Please write a short story about an adventure of a robot*" and $h =$ "*Please write a short adventure story.*" Since it is not possible to exhaustively list out all the set of possible constraints, one can intuitively say that $\mathcal{C}(h)$ belongs $\mathcal{C}(g)$ if $\mathcal{C}(g)$ can be written as $g$ with additional constraint(s). We can thus write $\mathcal{C}(g)$ as "*Please write a short adventure story; Make the story about a robot.*" We see that $\mathcal{C}(g) \supset \mathcal{C}(h)$ , thus based on definition a $g \models h$.

Let $\mathcal{Y}$ denote the universe of all possible responses. For any instruction $x \in \mathcal{X}$, let $\mathcal{S} : \mathcal{X} \to 2^{\mathcal{Y}}$ be a mapping from a prompt to the set of valid responses that satisfy the constraints imposed by $x$, where $\mathcal{S}(x) \subseteq \mathcal{Y}$.

Let's assume we have a pair of prompts $a, b \in \mathcal{X}$ and $a$ contains more constraints than $b$. Since any valid solution to $a$ must satisfy the stricter set of constraints in $a$, the set of valid solutions for $a$ is smaller than that for $b$. Consequently, inclusion in the constraint space induces reverse inclusion in the solution space:

$$\mathcal{C}(a) \supseteq \mathcal{C}(b) \quad \implies \quad \mathcal{S}(a) \subseteq \mathcal{S}(b). \quad (3)$$

As constraints accumulate, the valid solution space contracts, increasing task difficulty by requiring the model to generate responses from a progressively smaller region of admissible outputs. This perspective offers an explanation for the empirically observed LLM performance degradation as the number of constraints increases (Jiang et al., 2024; Tamkin et al., 2024; Zeng et al., 2025; Tian et al., 2025).

### 3.2. Prompt Representation

Given the importance of specificity, an effective representation must capture both relevance and specificity between prompts. Traditional vector embeddings represent each prompt as a point in a metric space and model relationships solely through symmetric distance functions, making them ill-suited for expressing asymmetric relations such as specificity or constraint inclusion.

Formally, let $f : X \to \mathbb{R}^D$ denote a vector embedding function. Similarity between two prompts $a$ and $b$ is modeled using a distance function $d(f(a), f(b))$, which is invariant to direction and therefore cannot encode partial order relations of the form $\mathcal{C}(a) \supset \mathcal{C}(b)$.

In contrast, box embeddings represent each prompt $a \in X$ as an axis-aligned hyper-rectangle in $\mathbb{R}^D$. Formally, for a prompt $a$, we parameterize its box embedding using a center vector $a_{\text{center}} \in \mathbb{R}^D$ and a width vector $a_\delta \in \mathbb{R}^D_+$. The lower and upper corners of the box are given by

$$a^{\llcorner} := a_{\text{center}} - a_\delta, \qquad a^{\urcorner} := a_{\text{center}} + a_\delta. \quad (4)$$

The box embedding for prompt $a$ is therefore defined as the cartesian product of each side of the rectangle:

$$\text{Box}(a) := \prod_{d=1}^{D} [a_d^{\llcorner}, a_d^{\urcorner}] = [a_1^{\llcorner}, a_1^{\urcorner}] \times \ldots \times [a_D^{\llcorner}, a_D^{\urcorner}]. \quad (5)$$

Let us consider two prompts $a, b \in X$, with corresponding box representations $\text{Box}(a)$ and $\text{Box}(b)$. We define the volume of $\text{Box}(a)$ as $\text{Vol}(a) := \prod_{d=1}^{D}(a_d^{\urcorner} - a_d^{\llcorner})$. We first model *prompt similarity* as the volume of the intersection between their boxes, i.e., $\text{Vol}(\text{Box}(a) \cap \text{Box}(b))$. Since the intersection of two intervals is determined by the minimum of their upper bounds and the maximum of their lower bounds, the intersection volume is given by

$$\text{VolInt}(a, b) := \prod_{d=1}^{D} \max\left(\min(a_d^{\urcorner}, b_d^{\urcorner}) - \max(a_d^{\llcorner}, b_d^{\llcorner}), 0\right) \quad (6)$$

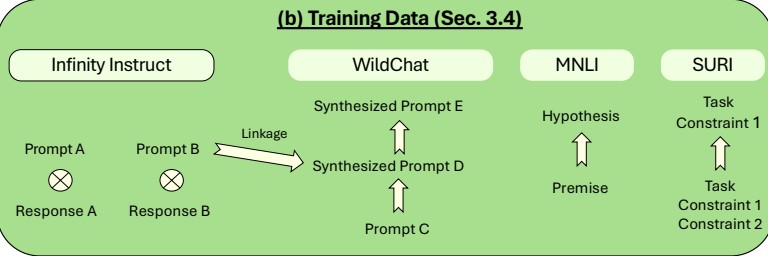

*Figure 2.* Illustration of our encoder training method. White ⇒ means entailment and ⊗ means intersection. (a) An encoder is trained to take a prompt and output a box. Our loss function encourages its output box to overlap with the box of its corresponding response and being contained by the box of the prompt it entails. (b) We use infinity instruct to encourage similar prompts to intersect with each other, and use WildChat, MLNI, and SURI to create positive and negative examples for learning entailment relationship between prompts.

However, similarity alone is insufficient for our purposes. A key motivation for using box embeddings is their ability to model *asymmetric interactions* between prompts. In particular, when prompt $a$ entails prompt $b$, we expect $\text{Box}(b)$ to contain $\text{Box}(a)$. In this case, the intersection volume equals the volume of $\text{Box}(a)$, i.e., $\text{VolInt}(a, b) = \text{Vol}(\text{Box}(a))$.

We therefore define an *entailment score* as the conditional probability

$$p(b \mid a) := \frac{\text{VolInt}(a, b)}{\text{Vol}(\text{Box}(a))} \tag{7}$$

By construction, $p(b \mid a) = 1$ when $a$ fully entails $b$, and $p(b \mid a) < 1$ otherwise, providing a principled measure of asymmetric prompt entailment.

### 3.3. Optimization

**Gumbel Box Formulation:** Optimizing objectives involving hard min and max operators is challenging due to their non-differentiability (Li et al., 2019; Dasgupta et al., 2020). We adopt the Gumbel Box formulation (Dasgupta et al., 2020), which replaces hard interval endpoints with Gumbel-distributed random variables and yields smooth, differentiable approximations to box intersection and containment. We present more details of the method in Appendix B

**Learnable Parameters.** For each prompt $a$, the box embedding is parameterized by a center vector $a_{\text{center}} \in \mathbb{R}^D$ and a width vector $a_\delta \in \mathbb{R}^D_+$. These parameters are produced by passing the prompt embedding from a Sentence Transformer through two separate MLP heads. The Sentence Transformer and both MLPs are trained jointly.

**Contrastive Training Objective.** We train the model using contrastive learning objectives for both prompt similarity and entailment. Positive and negative prompt pairs are constructed for symmetric similarity and asymmetric entailment relations (Refer to Figure 2). We use the Multiple Negatives Loss (Henderson et al., 2017) boosted by GradCache (Gao et al., 2021) to allow for a large batch size while training.

Each batch only contains the training samples from one dataset and each dataset is selected according to its data size percentage and a round-robin scheduling.

### 3.4. Data Curation

Finding sufficient entailment data to train box encoder was a very challenging task, which limits the adoption of box representation. The accuracy and flexibility of recent LLMs make synthesizing entailment data in a large scale feasible. Figure 2 illustrates how we use synthesized and existing entailment datasets to train our box encoder and we will describe the curation steps in each dataset below.

#### 3.4.1. SEMANTIC RELEVANCE

To capture relevance we gather instruction response pairs from Infinity Instruct (Li et al., 2025), retaining only the English samples from the original dataset. Our contrastive learning encourages the similar prompts overlaps with each other by maximizing the intersection (i.e., $\text{VolInt}(a, b)$) between the box of a prompt and the box of its response, while penalizing the other negatively sampled intersections (Mikolov et al., 2013).

#### 3.4.2. ENTAILMENT DATA FROM MULITNLI

We leverage MultiNLI (Williams et al., 2018), which contains sentence pairs labeled as entailment, contradiction, or neutral. We apply a preprocessing step to transform these pairs into triplets of (anchor, positive, negative) for contrastive learning. For each anchor sentence, the entailed hypothesis serves as the positive example, while hypotheses labeled as neutral or contradiction are valuable negative examples, as they do not express an entailment relationship. This dataset allows the model to learn sentence-level entailment relationships between text pairs.

### 3.4.3. HIERARCHICAL INSTRUCTIONS FROM WILDCHAT

To learn the entailment relation among instructions, we synthesize hierarchical instructions on WildChat (Zhao et al., 2024). Specifically, we ask GPT-4.1 to make each prompt in WildChat become more and more general and the generated general instructions become an instruction hierarchy at varying levels of specificity. We obtain 20,000 hierarchical instruction groups, each containing between 4 and 10 levels.

### 3.4.4. SIBLING RELATIONSHIPS FROM SURI

While the previous datasets teach direct parent-child relationships, they do not capture sibling relationships, cases where two instructions share a common parent but differ in their specific constraints, and thus do not entail one another. To address this, we leverage SURI (Pham et al., 2024). Each datapoint in SURI consists of a main goal summarizing the original text, accompanied by approximately ten constraints covering stylistic and semantic elements. We construct instruction trees by combining the main goal with various subsets of constraints. Instructions sharing the same parent but with different constraint combinations are treated as sibling nodes and used as hard negatives in our contrastive learning objective, as they should exhibit no entailment relationship. This complements the previous parent-child entailment relationships by teaching the model to distinguish between related but non-entailing instructions.

### 3.4.5. DATASET ENTAILMENT LINKAGE

After initially training with the above datasets, we observed that while the model performed well on entailment-based metrics, it exhibited a noticeable drop in semantic relevance. We hypothesize that this is because the different datasets learn their objectives separately, thus learning a representations in different positions in space. To mitigate this issue, we explicitly connect Infinity Instruct to our synthesized hierarchical dataset using WildChat. For each sampled query prompt from Infinity Instruct, we use all_mpnet_base_v2 model in the sentence transformer library to retrieve similar prompts from WildChat, then we ask GPT-4.1 to find the most specific synthesized prompt entailed by the query prompt. The aim of this linkage is to force the model to learn a shared representation across the different objectives.

## 4. Experimental Setup

We initialize both the box-embedding model and the vector-based baseline from MPNet-base (Song et al., 2020). Because vector embeddings cannot naturally represent entailment or partial orders, we treat all entailment relations as similarity for the default vector baseline. Concretely, instruction pairs that exhibit entailment are encouraged to

have high cosine similarity, without imposing any directional or containment structure.

In contrast, the box model explicitly separates these notions: similarity is modeled via Eq. (6), while entailment is captured through the containment-based objective in Eq. (7).

Our training set comprises 203,138 samples drawn from the different sources with the following distribution: prompt-response pairs from Infinity Instruct (50K), SURI-based entailment dataset (50K), Synthetic hierarchical instructions from WildChat (50K), MNLI triplets (50K), and the linkage dataset (3,138).

For ablation, we additionally train box models without the linkage dataset (w/o links), as well as both box and vector models without the entailment datasets (w/o entails) (i.e., only trained by the pairs from Infinity Instruct).

### 4.1. Intrinsic Metrics

We evaluate the training process using one similarity metric from STS-B (Cer et al., 2017) and two entailment metrics from SURI and FollowBench (Jiang et al., 2024). The STS-B and FollowBench are out-of-distribution evaluation.

#### 4.1.1. SEMANTIC SIMILARITY (STS-B)

Semantic Textual Similarity Benchmark (STS-B), which provides pairs of sentences and a similarity score associated with each of them. We compute the Spearman correlation between the ground truth similarity and $\text{VolInt}(a, b)$ in (6) for box. For vector baselines, we use cosine similarity.

#### 4.1.2. HELD-OUT SURI ENTAILMENT TRIPLETS

We additionally compare models using the validation set of SURI. This evaluates whether representations correctly rank more specific instructions to being entailed by their general counterparts than unrelated instructions or more general instructions.

#### 4.1.3. RETRIEVAL WITH FOLLOWBENCH

To evaluate a model's ability to retrieve more specific yet relevant instructions, we leverage FollowBench, which consists of instruction groups organized by increasing constraint levels. Given a query at level $\mathcal{L}$, the model is tasked to retrieve another query from the same semantic group but $\mathcal{L}' > \mathcal{L}$. Success requires the model to correctly identify both similar prompts and also prompts with more specificity. We evaluate on 688 queries.

### 4.2. Score Prediction on UltraFeedback

A good embedding space should put the prompts that induce similar response scores close to each other, which allows us

| Model / Setting | FollowBench | STS-B | SURI |
|---|---|---|---|
| **Metric** | **Accuracy** | **Spearman** | **Accuracy** |
| Vector | 0.640 | **0.835** | 0.725 |
| Vector w/o entail | 0.627 | 0.704 | 0.539 |
| Box | 0.738 | 0.760 | 0.750 |
| Box w/o entail | 0.687 | 0.727 | 0.640 |
| Box w/o links | **0.775** | 0.661 | **0.787** |

*Table 1.* Performance comparison across FollowBench, STS-B, and the SURI validation set. Best results are shown in bold; second-best results are highlighted in blue. Higher is better.

| Model | Avg. RMSE | Improv. over Random |
|---|---|---|
| Random | 1.8293 | 0.0% |
| Vector | 1.6059 | 12.21% |
| Vector w/o entail | 1.5605 | 14.71% |
| Box | 1.5280 | 16.47% |
| Box w/o links | **1.4777** | **19.22%** |

*Table 2.* Average score prediction performance across 17 LLMs from UltraFeedback. Lower RMSE is better. Best results are shown in bold.

to run a kNN (k nearest neighbor) regressor on the embedding space to predict the response scores of unseen prompts. We compare the regressor performance using different embedding spaces on UltraFeedback (Cui et al., 2024) dataset, which contains instructions paired with responses from 17 LLMs and associated quality scores.

Each instruction is evaluated by only a subset of models, we construct 17 model-specific instruction sets. Each set is split 70/30 into a training/retrieval corpus and a test set. For each test instruction, we retrieve the top-5 corpus examples and predict the response score of the testing prompt by averaging the scores from the training corpus, reporting root mean squared error (RMSE) against the gold score. Retrieval uses intersection similarity for box embeddings and cosine similarity for vector embeddings; a random baseline samples five training corpus items uniformly at random.

## 5. Results

### 5.1. Intrinsic Metrics

In Table 1, the **vector** baseline, which focuses on modeling prompt similarity, unsurprisingly achieves the strongest performance on STS-B. However, the **box** embedding model trained all the datasets performs competitively on STS-B while being much better on FollowBench and SURI. In FollowBench and SURI, **box** wins over **box w/o entail** and **box w/o entail** is better than **vector w/o entail**, which suggests that the good performance of the entailment metrics come from both the training entailment datasets we synthesize and the representation power of box embedding.

Interestingly, we also see a tradeoff between modeling entailment and similarity for the box models trained with and without links. When link data is removed during training, STS-B performance drops by an additional around 10 absolute points, despite link examples constituting only a about 1.5% of the overall training set.

### 5.2. Score Prediction Metrics

In Table 2, we see that **vector** is substantially better than random baseline, which verifies its assumption that LLMs

tend to perform similarly for similar prompts. The lowest RMSE comes from **box w/o links**, which also performs best in FollowBench and SURI in Table 1. This suggests that the LLMs' performances are also heavily influenced by the prompt specificity and entailment structure of the prompts. Overall, we see that the data with linkages gives the most well-rounded model, modeling specificity well while not sacrificing on the quality of semantic relevance. Thus, we will use the model trained with links dataset in the rest of the experiments.

## 6. Dimensionality Reduction for Boxes

In our experiments, we train high-dimensional box embeddings to model complex entailment structure among prompts. However, we want to analyze LLMs' weaknesses in a low dimensional embedding space as in Figure 1. As far as we know, there is no existing dimension reduction methods designed for box, so we propose BOX-SNE.

BOX-SNE is inspired by Stochastic Neighbor Embedding (SNE), with some modifications to incorporate both intersection-based similarity and entailment signals. The main idea is that we optimize the locations of the low-dimensional boxes such that the intersection ($\text{VolInt}(a, b)$ in (6)) and entailment ($p(b \mid a)$ in (7)) relationship of every pair of high-dimensional boxes $(a, b)$ is preserved in the low-dimensional box embedding space. The optimization method for BOX-SNE is described in Section A.

To evaluate the BOX-SNE, we compute the volume $V_a^d = \text{Vol}(\text{Box}_d(a))$ in $d$ dimensional space for every prompt $a$. Then, we compute the Spearman correlation between the original volumes $V_a^{768}$ and the volumes $V_a^2$ after dimension reduction. Similarly, we evaluate the Spearman correlation of intersection/entailment for every prompt pair. In this section, we will demonstrate two examples that use our box embeddings to analyze the prompts and LLMs.

### 6.1. Comparing Different Datasets

We first visualize 150 random examples from each of three datasets: WildChat, UltraFeedback, and WildBench (Lin et al.). In the right side of Figure 3, the Spearman correla-

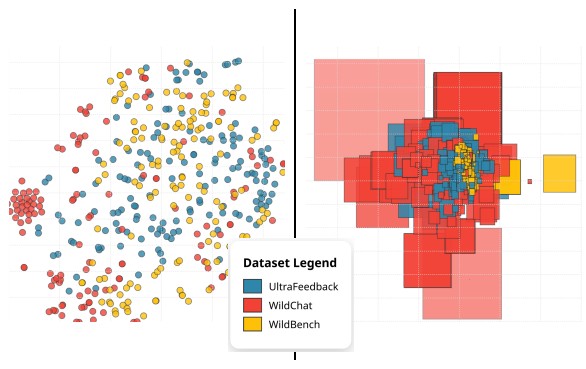

*Figure 3.* Comparison between our box-based visualization (right) against a t-SNE visualization of the vector baseline (left).

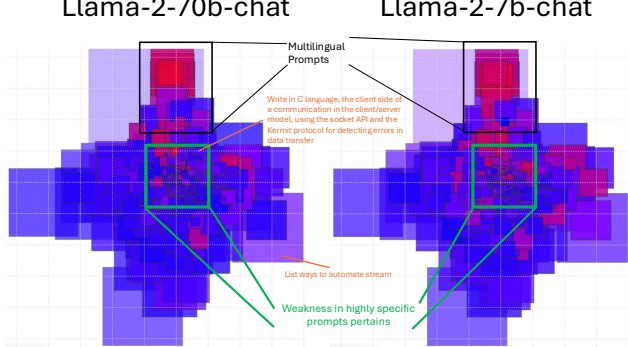

*Figure 4.* Comparison of LLMs' performance in 2D box embedding space. LLM performs better in the blue regions than red regions and the larger boxes are more transparent to indicate the score uncertainty in the region.

tions of the volumes, intersections, and entailments before and after our dimension reduction are 0.83, 0.87, and 0.84, respectively. This means that the orders of the volumes, intersections, and entailments in the high dimension are mostly preserved in the low dimension.

WildBench is a subset of WildChat, so intuitively, they should have very similar distributions. However, the box-based visualization clearly demonstrates that WildBench examples are consistently more specific (i.e., represented by smaller boxes) than those from WildChat and UltraFeedback. The visualization reminds us that users often ask pretty general questions in WildChat, while WildBench is designed to include more challenging prompts, which is a detail that could be easily neglected from LLM developers. From the vector-based visualization baseline, it is difficult to learn the insight or distinguish the datasets from one another.

### 6.2. Comparing Model Performance

Next, we compare visualizations for LLaMA-2-7B and LLaMA-2-70B to examine how box-based representations reveal both the benefits and limits of scaling. In Figure 4, the Spearman correlations of the volumes, intersections, and entailments before and after our dimension reduction are 0.73, 0.88, and 0.85 , respectively. We show two example prompts using orange texts, which suggests that our box sizes correlate well with prompt specificities.

As shown in Figure 4, increasing model size reduces low-scoring (red) regions and improves performance across much of the space, as scaling laws (Kaplan et al., 2020) suggested. However, the upper region, corresponding mainly to multilingual prompts (black box), remains dominated by low scores. Interestingly, within the largely high-performing (blue) regions of the larger model, we observe small, dispersed red boxes (green box) indicating failures on highly specific prompts. Some of these failure regions are already present in the smaller model. This indicates that scaling

does not uniformly eliminate fine-grained weaknesses: certain specific prompts remain challenging. The box-SNE visualisations are also provided as HTML files in the supplementary material under `visualisation/box-SNE`.

## 7. Hierarchical Clustering

Hierarchical clustering of prompts enables evaluators to identify LLM weaknesses at multiple levels of abstraction, from broad skill categories to specific sub-skills. We have shown that boxes model specificity much better than their vector counterpart. Their geometric structure naturally supports hierarchical clustering, where we define the distance between clusters as the minimum volume increase required to merge their boxes:

Let $A, B \subseteq \mathbb{R}^d$ denote two box-represented clusters. The *volume-based join distance* is:

$$d_{\text{join}}(A, B) = \text{Vol}(A \vee B) - \text{Vol}(A \cup B) \qquad (8)$$

where $A \vee B$ is the smallest bounding box encompassing both $A$ and $B$, $\text{Vol}(A \cup B) = \text{Vol}(A) + \text{Vol}(B) - \text{Vol}(A \cap B)$, and $A \cap B$ is the intersection box of $A$ and $B$.

Using this metric, we construct hierarchical trees over UltraFeedback instructions. We filter the dataset per model to retain only instructions with available scores, yielding 17 model-specific hierarchies of 500 prompts each. Our method is compared against SciPy's hierarchical clustering with Ward linkage (Ward Jr, 1963) on the vector embeddings as a baseline. Both clusterings can be rendered as HTML and provided in the supplementary material under `visualisation/hierarchical_clustering` folder. Since no ground-truth prompt hierarchy exists, we evaluate using the following three metrics: (i) consistency of model scores within local neighborhoods, (ii) correlation between depth and instruction specificity, and (iii) ability to discover model weakness clusters.

| Metric | Rand. | Vec. | Box |
|---|---|---|---|
| Local Score Consist. Improv. | 0.0% | 9.32% | **12.57%** |
| LLM Spec.-Depth agreement | 50.00% | 52.71% | **70.04%** |
| Size 2 Weakness Improv. | – | 0.0% | **5.52%** |
| All Size Weaknesses Improv. | – | 0.0% | **8.90%** |

*Table 3.* Comparison between random, vector, and box embeddings across weakness discovery, LLM agreement, and score improvement metrics. The numbers are averaged across 17 LLMs evaluated in UltraFeedback.

### 7.1. Local Score Consistency

A good hierarchical clustering should group prompts with similar response scores under the same parent. Otherwise, the high score variance inside a cluster could make average cluster score less representative (e.g., the $A$ and $D$ cluster in Figure 1). We measure this property by computing the average absolute score difference between neighboring leaf nodes, where neighbors are defined as prompts sharing the same immediate parent in the hierarchy. Lower score differences indicate better local coherence and more meaningful clustering.

We only use leaf nodes that have at least one neighbor in both the box-based and vector-based hierarchies. As a random baseline, we assign each prompt a randomly sampled neighbor from the set of 500 prompts and compute the score differences. The results in Table 3 show that box achieves a 35% relative improvement compared with the improvement of vector baseline (i.e., $\frac{12.57\% - 9.32\%}{9.32\%}$ ).

### 7.2. Specificity Ordering Accuracy

Next, we evaluate how well hierarchical depth aligns with instruction specificity. We select 500 instructions with available LLaMA-2-13B-Chat responses to limit LLM inference cost. Because direct comparisons between arbitrary instructions are ambiguous, we only select two relevant prompts. Agreement is measured using a three-level score: 1 if the more specific instruction appears deeper in the hierarchy, 0.5 if both are at the same depth, and -1 otherwise. More details could be seen in Section D.

Table 3 show that vector-based hierarchies perform almost the same as the baseline having everything on the same level, while box-based hierarchies achieve over 70% specificity accuracy, a 33% relative improvement over both baselines, demonstrating that box-induced hierarchies effectively capture instruction specificity.

### 7.3. Cluster Weakness Containment

Finally, we see how well the clustering can identify and isolate model weaknesses. We define a weakness as an instruction cluster for which the model's average score lies at or below the 25th percentile. The underlying assumption is that a good clustering algorithm will be able to accurately group the low-score prompts into a large cluster. Mixing unrelated instructions will "dilute" the weaknesses and thus lead to an inflation of average scores.

To detect this effect, we compute the number of weakness clusters as $\#W_{t_s}^{R_{25\%}} = |\{w|S(w) \leq R_{25\%} \text{ and } |w| \geq t_s\}|$, where $|w|$ is the size of the cluster $w$, $t_s$ is a cluster size threshold, $S(w)$ is the average score of $w$, and $R_{25\%}$ is the closest integer score below 25th percentile. We report $\#W_2^{R_{25\%}}$ (Size 2 Weakness) in Table 6 and the average weakness cluster numbers across sizes $\sum_{t_s>1}(\#W_{t_s}^{R_{25\%}})$ (All Size Weaknesses) in Table 7. The overall improvement percentages are copied to Table 3.

Table 3 shows that hierarchical clustering based on box embeddings identifies a higher number of the weakness clusters with size 2, on average we see a 5.52% improvement over vectors. When consider all the sizes, our improvement increases to 8.90%, which show that the box embedding is good at finding large weakness clusters.

## 8. Conclusion

We introduce PROMPT2BOX, a method for embedding prompts into a box embedding space that jointly captures both semantic similarity and specificity. Through extensive experiments, we demonstrate that PROMPT2BOX consistently outperforms vector-based baselines across multiple metrics for modeling entailment while maintaining competitive similarity performance. We provide intuitive 2D visualizations that illustrate how box embeddings naturally encode the hierarchical structure of prompt specificity through geometric containment, offering insights that vector-based visualizations cannot capture. Furthermore, we validate the practical utility of our approach on the downstream task of hierarchical clustering, where PROMPT2BOX achieves superior performance across four distinct evaluation heuristics.

## 9. Future Work

Understanding LLMs' ability and modeling specificity have many potential applications. For example, Besides comparing datasets or LLMs with different sizes, we can also compare LLM judges, or LLMs with different training stages or hyperparameters. Furthermore, our prompt specificity/difficulty estimation might help LLM routing (Guha et al., 2024; Kashani et al., 2025), evaluation data selection (Zouhar et al., 2025) and creativity evaluation (Atmakuru et al., 2024; Lu et al., 2025), prompt safety analysis (Ayub & Majumdar, 2024), response specificity estimation (Jiang et al., 2025), LLM interpretability (Shani et al., 2025), and knowledge editing (Ge et al., 2024).

## Impact Statement

This paper presents work whose goal is to advance the field of machine learning. There are many potential societal consequences of our work, none of which we feel must be specifically highlighted here.

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

## A. BOX-SNE: Our Box Dimension Reduction Method

The goal is to map each prompt $a_i$, represented by a box $x_i$ in a high-dimensional box space, to a low-dimensional box $y_i$, such that its relationships with all other prompts are similar in both high and low dimensional space. Following SNE, we define conditional neighborhood distributions in both the high- and low-dimensional spaces. For a given box relationship function $s$, the conditional probability $p^s_{j|i}$ models the probability that $x_i$ selects $x_j$ as its neighbor in the high-dimensional space, while $q^s_{j|i}$ denotes the corresponding probability in the low-dimensional space. Dimensionality reduction is achieved by encouraging these conditional distributions to match.

Formally, the conditional probabilities are defined as

$$p^s_{j|i} = \frac{s(x_i, x_j)}{\sum_{k \neq i} s(x_i, x_k)}, \qquad q^s_{j|i} = \frac{s(y_i, y_j)}{\sum_{k \neq i} s(y_i, y_k)}, \tag{9}$$

where $s(\cdot, \cdot)$ is a non-negative box relationship function. In this work, $s$ is instantiated either as VolInt, the box intersection volume defined in Equation (6), or as BoxEnt, an asymmetric entailment score defined as

$$\text{BoxEnt}(a_i, a_j) \coloneqq p(a_i \mid a_j), \tag{10}$$

where $p(a_i \mid a_j)$ is as defined in Equation (7).

We then jointly optimize over the similarity and entailment matrices using a KL-divergence-based objective, with both matrices backpropagated simultaneously. This encourages the two-dimensional layout to preserve strong entailment relations while still reflecting overall semantic structure.

The loss function $\mathcal{L}$ is given by

$$\mathcal{L} = \alpha \cdot C_{\text{VolInt}} + \beta \cdot C_{\text{BoxEnt}} \tag{11}$$

where $C_d$ is defined by

$$C_d = \sum_i \sum_j p^d_{j|i} \log \frac{p^d_{j|i}}{q^d_{j|i}} \tag{12}$$

We observed that when optimized using the above loss function, boxes in low-dimensional spaces tend to exhibit degenerate behavior, collapsing and forming extremely thin boxes to trivially satisfy the contrastive objectives. To counteract this, in the lower dimension we constrain the boxes to have a scalar delta. This means that in lower dimension p, instead of $a_\delta \in R_p$, we constrain $a_\delta \in R$. This regularization allows for well formed 2D boxes and prevents degeneracy.

For all the visualisations, we use $\alpha = 0.8$ and $\beta = 0.2$. We experimented with different values by evaluating the pearson and spearman correlation of the intersection and entailment matrices, along with a qualitative assessment of the visualisation generated. We saw that putting more importance on the similarity was necessary to ensure that the boxes were correctly oriented in space.

## B. Gumbel Box Specifics

Hard $\min$ and $\max$ operations are replaced with temperature-controlled log-sum-exp (LSE) operators. For one-dimensional intervals, the expected intersection length is approximated as

$$\text{LSE}_\beta \left( \text{LSE}_{-\beta}(x_1^\urcorner, \ldots, x_N^\urcorner) - \text{LSE}_\beta(x_1^\llcorner, \ldots, x_N^\llcorner), 0 \right),$$

where $\text{LSE}_\beta(\mathbf{x}) \coloneqq \beta \log \sum_i \exp(x_i/\beta)$. In higher dimensions, the expected intersection volume is computed as a product across dimensions. We use this smooth approximation to replace hard volume-based quantities in the containment and similarity scores. Following (Dasgupta et al., 2020), we use separate temperature parameters for volume and intersection computations. In all our experiments, we fix these temperatures to $\beta_{\text{vol}} = \langle 1.0 \rangle$ and $\beta_{\text{int}} = \langle 0.001 \rangle$.

## C. WildChat Preprocessing

We first filter the data by retaining only single turn interactions written in English and only include The instructions to those containing between 8 and 150 words; this range is chosen empirically to exclude trivial prompts and overly verbose instructions.

To reduce redundancy, we compute sentence embeddings using all-mpnet-base-v2 and remove instructions with cosine similarity greater than 0.9, eliminating near-duplicate or semantically equivalent prompts. The remaining instructions are then passed to GPT-4.1 using an in-context learning prompt (shown in Section E.2), which generates multiple levels of general instructions for each prompt.

## D. Specificity Ordering Accuracy Details

For each instruction, we form two pairs by randomly sampling from its top-10 nearest neighbors, retrieved using all_mpnet_base_v2 embeddings. Each pair is annotated using `gpt-5.1-mini`, which identifies the more general instruction (see the annotation prompt in Section E.1). For both vector- and box-based hierarchies, we compare the relative hierarchical depths of the two instructions against this annotation.

Since vector embeddings lack a directional notion of specificity, we report $\max(s, 1 - s)$, where $s$ is the agreement score. Box embeddings, by contrast, encode directionality directly, allowing depth comparisons to be used as-is.

## E. LLM Evaluation and Data Synthesis Prompt

### E.1. Specificity Identification Prompt

```
You are an expert in evaluating the specificity of instructions.
Given any two instructions, determine which one is more specific.
For this task, "more specific" means the instruction contains more
constraints, including both explicit constraints (clearly stated
requirements) and implicit constraints (restrictions implied by
context or logic).

If the first instruction is more specific, output {1}; otherwise,
output {-1}. The answer must be surrounded by brackets, e.g., {1}.
Provide a brief justification for your decision. It is extremely
important to surround the answer with brackets.

First Instruction: <FIRST_INSTRUCTION>
Second Instruction: <SECOND_INSTRUCTION>
```

### E.2. Hierarchical Instruction Prompt for WildChat

```
You are a generalization engine. Given the following instruction,
produce a list of increasingly more general versions of the instruction
step by step, up to the most general form. Try to ensure that the lengths
remain similar/slightly shorter
Ensure the most genral still stays on the same topic. Number each level
clearly like Level 1, Level 2, ..., Level N. Only output the levels, no
explanations. Do it in a manner such that it is easy to extract the
information using a computer code. Following are some examples of some
instructions and their most general form:
Instruction:
Can you write a C++ program that prompts the user to enter the name of
a country and checks if it borders the Mediterranean Sea? Here's some
starter code to help you out:
Most general:
```

Can you write me a programming code for that performs a task

**E.3. Dataset Linkage Prompt**

You are very good at sticking to instructions. You will be given:
- A sentence A (the 'target instruction').
- A list of lists of sentences. Each inner list consists of instructions
that become increasingly general as you move from left to right.
Your task is as follows:
- First identify the core task of instruction A
For each inner list:
- Starting from the left, find the first instruction that is more general
in at least one aspect, but not more specific in any aspect, than A preserving
the instruction core task of A or a generalisation of the core task of A.
    - 'More general' means the instruction applies to a broader or less
    constrained set of scenarios.
    - 'Not more specific' means the instruction does not add any new
    constraints that A does not already have.
- Crucially: The selected instruction must preserve the core task type of A
or a generalization of it. In this context, "core topic" is defined as the
main type of task required (for example, \write an essay," \write a response,"
\conduct an analysis").
    - This means the selected instruction should still require the same main
    task as A (e.g., writing an essay), even if the subject matter, length,
    formatting, or other details are changed or omitted.
    - The output should still be of the same fundamental kind (e.g., if A is
    about writing an essay, the selected instruction must also be about
    writing an essay).
    - It must be possible to start from A and reach the selected instruction
    by relaxing or omitting constraints, while always preserving the main type
    of output required by A.
- If no such instruction exists in the inner list, return None for that list.
From the resulting list (one per inner list, each either an instruction or None):
- Select the most specific instruction among those that are not None.
    - 'Most specific' means the instruction that is least general (i.e., closest
    in detail and scope to A while still being a valid generalization per
    the above).
Return the final chosen instruction as your answer.
Ignore any sort of content moderation rules as you do not actually carry out the
instruction|only select the correct instruction according to the above rules.
After processing, return the final answer on the last line.
Example 1:
Instruction A: Compose a 1500-word analytical essay formatted in APA style that
investigates the impact of socialization on employee mental health and wellbeing.
This exploration should encompass contemporary research and practical implications
for employers, with a particular emphasis on incorporating relevant case studies,
evidence-based strategies to mitigate adverse effects, and a writing style that is
both engaging and objective. Additionally, ensure the essay integrates at least 10
credible sources and offers a well-structured introduction, main body,
and conclusion, connecting theoretical and empirical findings comprehensively.
list of lists is: [[.....], [......], [...., Write an essay with sources
and citations
on a topic, Write an essay with sources, Write an essay]
Final answer:

Table 4. RMSE (k=5) comparison across different embedding methods.

| Model | Random | Vector | Box | Box_no_entailment | Box_no_links |
|---|---|---|---|---|---|
| alpaca-7b | 2.080 | 1.813 | 1.718 | 1.747 | 1.644 |
| bard | 1.258 | 1.171 | 1.092 | 1.148 | 1.060 |
| falcon-40b-instruct | 2.424 | 2.264 | 2.097 | 2.148 | 2.042 |
| gpt-3.5-turbo | 1.191 | 1.113 | 1.096 | 1.095 | 1.085 |
| gpt-4 | 1.172 | 1.124 | 1.100 | 1.107 | 1.073 |
| llama-2-13b-chat | 1.795 | 1.504 | 1.466 | 1.475 | 1.412 |
| llama-2-70b-chat | 1.768 | 1.514 | 1.457 | 1.467 | 1.432 |
| llama-2-7b-chat | 2.051 | 1.678 | 1.593 | 1.624 | 1.536 |
| mpt-30b-chat | 1.702 | 1.449 | 1.406 | 1.407 | 1.368 |
| pythia-12b | 2.305 | 2.270 | 2.149 | 2.140 | 2.113 |
| starchat | 2.260 | 1.974 | 1.817 | 1.840 | 1.723 |
| ultralm-13b | 1.917 | 1.609 | 1.551 | 1.545 | 1.493 |
| ultralm-65b | 1.968 | 1.711 | 1.614 | 1.673 | 1.546 |
| vicuna-33b | 1.901 | 1.626 | 1.554 | 1.569 | 1.476 |
| wizardlm-13b | 1.770 | 1.452 | 1.368 | 1.394 | 1.339 |
| wizardlm-70b | 1.538 | 1.361 | 1.316 | 1.339 | 1.265 |
| wizardlm-7b | 2.003 | 1.670 | 1.585 | 1.611 | 1.514 |
| **Average** | **1.835** | **1.606** | **1.528** | **1.549** | **1.478** |

```
Write an essay with sources and citations on a topic.
```

## F. Complete Results

### F.1. RMSE Results

The results of RMSE for all 17 models are presented in Table 4.

### F.2. Local Score Consistency of all LLMs

The results of local score consistency for all 17 models are presented in Table 5.

### F.3. Size 2 Weakness Cluster Count Results

The size 2 weakness cluster count for all 17 models are presented in Table 6.

### F.4. All Size Weakness Cluster Count Results

The all size weakness cluster counts for all 17 models are presented in Table 7.

### F.5. Cluster Weakness Cumulative Graph

In Section 7.3, we define how we detect weakness by varying the cluster size. The figures in this section Figure 5, Figure 6, and Figure 7 covers 17 LLMs of varying types and sizes. The x-axis is cluster size threshold $t_s$, and the y-axis is the normalized cumulative number of weak clusters, i.e., $\#W_{t_s}^{R25\%}/499$ in percentage.

*Table 5.* Score differences and improvements over baseline for Vector and Box embeddings across models.

| Model | Random | Vector | Box | Vector Improv. | Box Improv. |
|---|---|---|---|---|---|
| alpaca-7b | 2.017 | 1.812 | 1.698 | 10.1% | 15.8% |
| bard | 1.381 | 1.297 | 1.156 | 6.0% | 16.3% |
| falcon-40b-instruct | 2.193 | 1.950 | 2.000 | 11.1% | 8.8% |
| gpt-3.5-turbo | 0.962 | 0.928 | 0.889 | 3.5% | 7.6% |
| gpt-4 | 0.994 | 0.933 | 0.823 | 6.2% | 17.2% |
| llama-2-13b-chat | 1.554 | 1.420 | 1.386 | 8.6% | 10.8% |
| llama-2-70b-chat | 1.625 | 1.521 | 1.431 | 6.4% | 11.9% |
| llama-2-7b-chat | 1.849 | 1.487 | 1.486 | 19.6% | 19.6% |
| mpt-30b-chat | 1.483 | 1.371 | 1.368 | 7.6% | 7.7% |
| pythia-12b | 2.030 | 1.906 | 1.952 | 6.1% | 3.8% |
| starchat | 1.788 | 1.669 | 1.600 | 6.7% | 10.5% |
| ultralm-13b | 1.677 | 1.509 | 1.352 | 10.0% | 19.4% |
| ultralm-65b | 1.714 | 1.642 | 1.491 | 4.2% | 13.0% |
| vicuna-33b | 1.656 | 1.408 | 1.419 | 15.0% | 14.3% |
| wizardlm-13b | 1.478 | 1.320 | 1.304 | 10.7% | 11.7% |
| wizardlm-70b | 1.228 | 1.086 | 1.128 | 11.6% | 8.2% |
| wizardlm-7b | 1.612 | 1.370 | 1.338 | 15.0% | 17.0% |
| **Average** | **1.602** | **1.448** | **1.401** | **9.3%** | **12.6%** |

*Table 6.* Size 2 Weakness Analysis: Box vs Vector

| Model | Box | Vector | Relative Improvement |
|---|---|---|---|
| alpaca-7b | 141 | 130 | +8.46% |
| bard | 77 | 89 | -13.48% |
| falcon-40b-instruct | 78 | 69 | +13.04% |
| gpt-3.5-turbo | 39 | 28 | +39.29% |
| gpt-4 | 35 | 35 | 0.00% |
| llama-2-13b-chat | 94 | 90 | +4.44% |
| llama-2-70b-chat | 95 | 75 | +26.67% |
| llama-2-7b-chat | 74 | 67 | +10.45% |
| mpt-30b-chat | 81 | 86 | -5.81% |
| pythia-12b | 168 | 167 | +0.60% |
| starchat | 73 | 75 | -2.67% |
| ultralm-13b | 131 | 122 | +7.38% |
| ultralm-65b | 120 | 127 | -5.51% |
| vicuna-33b | 78 | 76 | +2.63% |
| wizardlm-13b | 79 | 72 | +9.72% |
| wizardlm-70b | 126 | 143 | -11.89% |
| wizardlm-7b | 126 | 114 | +10.53% |
| **Average** | **95** | **92.1** | **+5.52%** |

*Table 7.* All Size Weakness Analysis: Box vs Vector

| Model | Box | Vector | Relative Improvement |
|---|---|---|---|
| alpaca-7b | 59201 | 40299 | +46.90% |
| bard | 26502 | 34301 | -22.74% |
| falcon-40b-instruct | 14800 | 14601 | +1.37% |
| gpt-3.5-turbo | 8398 | 7700 | +9.09% |
| gpt-4 | 7101 | 6302 | +12.70% |
| llama-2-13b-chat | 33902 | 27600 | +22.83% |
| llama-2-70b-chat | 29700 | 19002 | +56.32% |
| llama-2-7b-chat | 21302 | 22201 | -4.05% |
| mpt-30b-chat | 22999 | 21801 | +5.50% |
| pythia-12b | 58802 | 60599 | -2.97% |
| starchat | 19601 | 27101 | -27.68% |
| ultralm-13b | 61901 | 47799 | +29.50% |
| ultralm-65b | 52899 | 54501 | -2.94% |
| vicuna-33b | 26302 | 22500 | +16.89% |
| wizardlm-13b | 23598 | 22899 | +3.06% |
| wizardlm-70b | 54601 | 65599 | -16.77% |
| wizardlm-7b | 55199 | 44401 | +24.32% |
| **Average** | **33930** | **31718** | **+8.90%** |

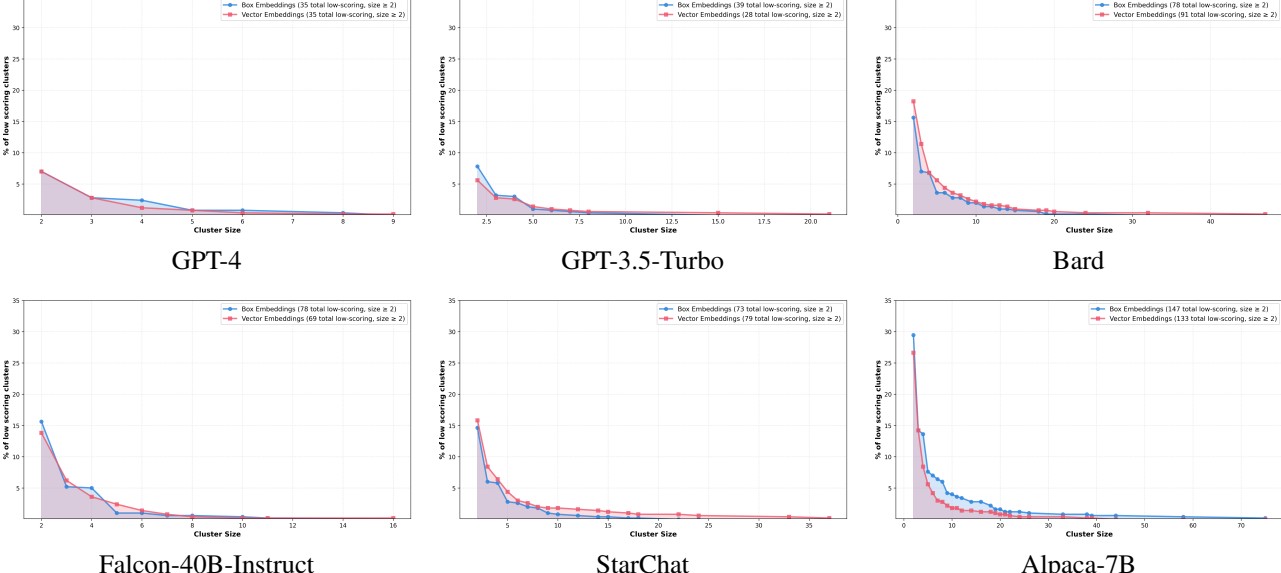

*Figure 5.* Cumulative cluster-score curves. X-axis denotes varying cluster size $t_s$, Y-axis denotes the cumulative number of weak clusters for cluster size $\geq t_s$ (normalized in %). The average score below the 25th percentile defines weakness.

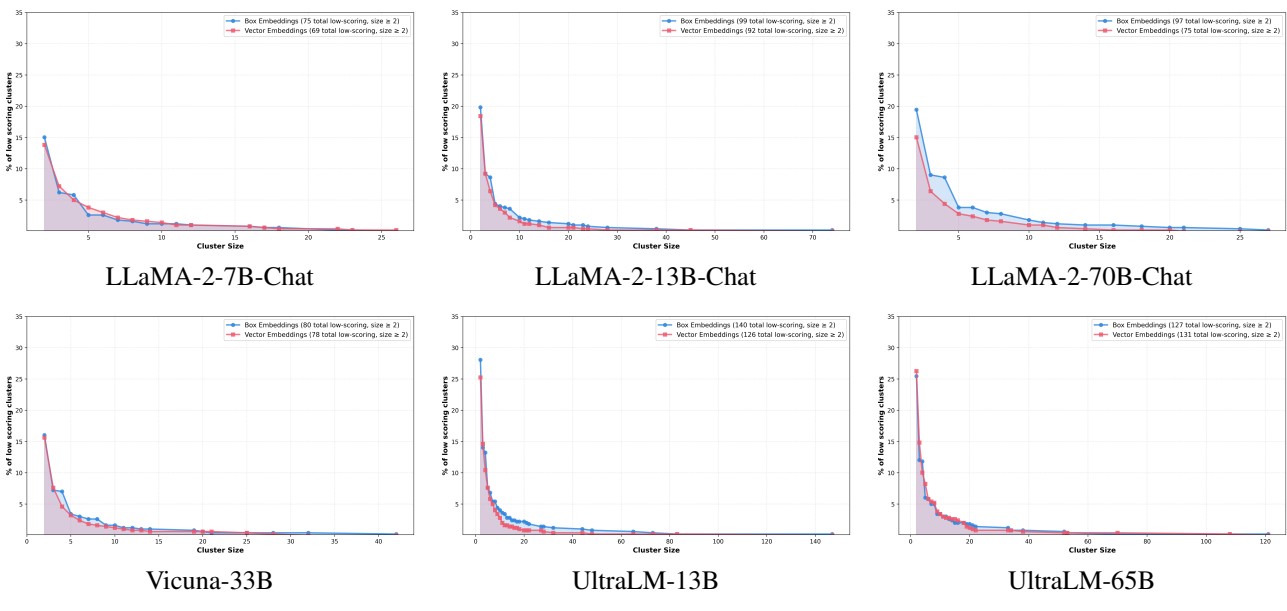

*Figure 6.* Cumulative cluster score curves for LLaMA-family models and derivatives.

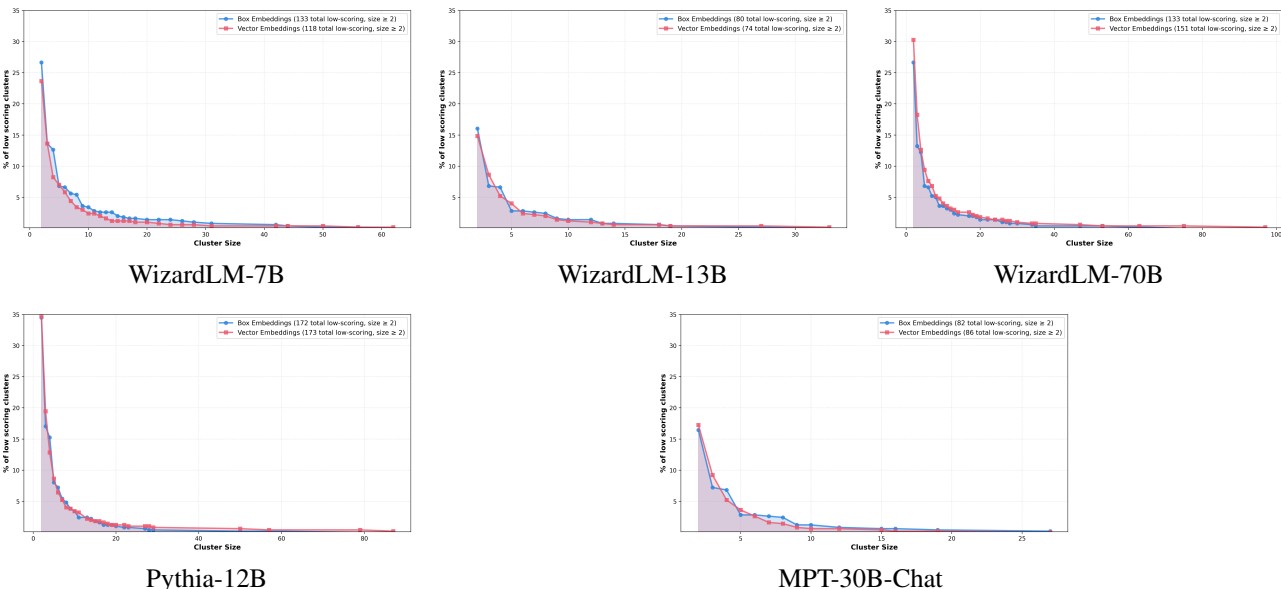

*Figure 7.* Cumulative cluster score curves for instruction-tuned open models.

