# OpenReview forum: "Prompt2Box: Uncovering Entailment Structure among LLM Prompts"
_ICML.cc/2026/Conference — Submitted to ICML 2026_

### Official Review · Reviewer_yYvq · 2026-03-11

**Soundness:** 3
**Presentation:** 3
**Significance:** 2
**Originality:** 1
**Overall Recommendation:** 3
**Confidence:** 3

**Summary:**

The paper proposes to identify weaknesses of LLMs by representing prompts using box embeddings, in which each prompt is associated with a hyper-rectangle in embedding space. The authors open their analysis by arguing that standard vector embeddings rely on symmetric distance functions which might fail to represent asymmetric relations, and propose to use box embeddings to solve this limitation. By switching to this representation, the authors aim to obtain better performance compared to standard embeddings on tasks involving asymmetric relationships between prompts, such as specificity, inclusion and entailment.
Finally, the paper also introduces a visualization technique (BOX-SNE) to project box embeddings into two dimensions for qualitative analysis.

**Compliance With Llm Reviewing Policy:**

Affirmed.

**Key Questions For Authors:**

- Would it be possible to explain better the claim regarding the superiority of Box embeddings over Poincaré and Hyperbolic Embeddings? Could the authors support this claim by citing known works in the literature or with specific experiments? Would it be possible to compare with state-of-the-art techniques for specificity or inclusion of prompts?
- Could the authors better clarify which technical novelties are being introduced in their work? As far as I understand, box embeddings and most of the methodological framework already exist in prior work, and it is not clear which contributions are new beside the visualization method.

**Limitations:**

Please see the Weaknesses and Questions sections above

**Strengths And Weaknesses:**

## Strengths
- Motivation and problem significance. Understanding the weaknesses of LLMs and identifying failure modes in prompt space is highly relevant for model evaluation and improvement of the training data.
- Presentation. The paper is overall well written, the motivation is explained clearly and the discussion of why symmetric similarity metrics may fail in specific settings is intuitive.
- Technically sound use of box embeddings. The use of box embeddings to represent hierarchical relationships is well grounded, and this representation seems a natural fit for modeling relations of specificity, inclusion and entailment.

## Weaknesses
- Limited technical novelty. The main methodological component of the paper, box embeddings, is already known in literature, and the biggest novel contribution appears to be the new visualization technique (BOX-SNE). The paper therefore appears to be the application of an existing embedding method to a new task. Such application is still very valuable and it is well supported and motivated, but the paper would benefit from a clearer explanation of what technical advances are being introduced.
- No comparison with Hyperbolic or Poincaré embeddings. The paper states that: "Box embeddings [...] have been shown to outperform other region-based representations such as [...] Poincaré Embeddings in modeling asymmetric relationships". However, this claim raises several concerns, since i) there are no citations of works supporting this general superiority of box embeddings over hyperbolic or Poincaré embeddings, and ii) despite this claim the paper does not include any comparison with hyperbolic methods, and the only evaluation is between standard vector embeddings and box embeddings. The absence of such comparisons makes it difficult to determine whether box embeddings are really preferable to hyperbolic embeddings in this setting.
- The evaluation scope appears narrow. The experimental evaluation primarily focuses on tasks that emphasize properties that box embeddings are specifically designed to capture. This is surely aligned with the paper's motivation, but it raises the concern that the evaluation may be biased toward the strengths of the proposed representation while overlooking other aspects. The paper would be strengthened by adding experiments to assess whether these box embeddings preserve the general semantic relationships captured by standard vector embeddings. The authors also report the scale of the training data: an analysis of how the number of training examples affects the performance of vector vs box embeddings would also strengthen the paper. Additionally, although the architecture of the encoder remains the same, a box embedding has a different number of parameters from a vector embedding (since a box embedding parameterizes each prompt using both a center and a width vector, it has double the number of parameters). This increased capacity may partly explain the improved performance on tasks related to hierarchy or entailment. The paper does not control for this difference, and therefore it is difficult to determine whether the observed gains arise from the modeling framework itself or simply from increased capacity. An analysis in this direction would significantly improve the work.

---

> ### Author Rebuttal · Authors · 2026-03-29
>
> We thank the reviewer for their response. We seek to answer major issues/complaints with the paper
>
> ## Novelty
> We believe our work is among the first to conceptualize prompts in this manner and to explicitly model prompt structure. From a methodological standpoint, our contribution also includes applying box embeddings to language modeling for the first time, and introducing a training framework that combines intersection-based loss with entailment loss. We also develop the data collection and data synthesis to support this kind of loss.  We introduce Box-SNE and incorporate a new formulation of hierarchical clustering based on boxes to analyze the learned structure.
>
> More broadly, we argue that novelty should not be restricted to entirely new techniques. Our work contributes through a combination of new perspective, formulation, and application. Prior influential works such as GPT-3[1] have similarly demonstrated that impactful contributions can arise from rethinking problem formulations and effectively integrating existing ideas. In this sense, we believe our approach offers sufficient novelty both in motivation and in execution. Our paper shows the relevance of box embeddings in modeling LLM weaknesses and potentially many more important tasks as we demonstrate in the future work section.
>
> ## Use of Box embeddings
>
> We adopt box embeddings based on prior work such as "Representing Joint Hierarchies with Box Embeddings"[2] and "Capacity and Bias of Learned Geometric Embeddings for Directed Graphs"[3], which demonstrate their effectiveness in modeling hierarchical and partially ordered structures over other methods like Poincare and hyperbolic structures. We will add these references in the paper. Our primary goal in this paper is not to argue that box embeddings are universally superior for modeling prompt space, but rather to highlight the importance of explicitly modeling prompt specificity.
>
> Box embeddings provide a natural mechanism to capture containment and hierarchy, making them a reasonable choice grounded in existing literature. While it is possible that alternative embedding methods could perform competitively or even better in this setting, conducting a comprehensive comparison across fundamentally different representation families is non-trivial and beyond the scope of this work. We leave such comparisons to future research.
>
> ## Evaluation scope
>
> We agree that while certain metrics are more directly aligned with measuring specificity, we also measure other metrics, such as RMSE, local neighbor score difference and number of weakness clusters, which are invariant to the model type. These metrics help demonstrate box embeddings provide a better representation space for prompts than vectors.
>
> We run an additional experiment where we compare the similarity between neighbors in the hierarchical clusters. and present the results in the table below. We use the all_mpnet_base_v2 model from huggingface as the embedding model and compute cosine similarity between neighbors. Even though the neighbor-based evaluation is inherently biased toward vector representations (as it relies on a vector similarity measure), our method achieves improved neighborhood quality while incurring only a modest (~16%) drop in semantic similarity.
>
>
> | Model/Metric                         | Vector  | Box      | Degradation |
> |--------------------------------|--------|----------| ----------- |
> | Local Vector Similarity   | **0.369**  | 0.310 | 16% |
>
>
> ## Number of parameters.
>
> We clarify that our model does not significantly increase parameter count. We use the same backbone as the vector baseline and simply add two lightweight linear projection heads to parameterize the center and offset. Each head consists of two linear layers with the hidden dimension of 768. This results in an increase of approximately 2 million parameters(768 x 768 x 2 x 2), which corresponds to roughly a 2% increase over the baseline model.
>
> We believe this additional capacity is minimal and unlikely to account for the observed performance gains. We will explicitly report this detail in the revised manuscript for clarity.
>
>
> [1] Brown and Mann and Ryder and Subbiah et al. "Language Models are Few-Shot Learners" (https://papers.nips.cc/paper/2020/hash/1457c0d6bfcb4967418bfb8ac142f64a-Abstract.html)
>
> [2] Patel and Dasgupta et al. "Representing Joint Hierarchies with Box Embeddings" (https://www.akbc.ws/2020/assets/pdfs/J246NSqR_l.pdf)
>
> [3] Boratko and Zhang et al. "Capacity and Bias of Learned Geometric Embeddings for Directed Graphs" (https://proceedings.neurips.cc/paper_files/paper/2021/hash/88d25099b103efd638163ecb40a55589-Abstract.html)

---

> > ### Author Rebuttal · Reviewer_yYvq · 2026-04-03
> >
> > I thank the authors for their response and clarifications. Below I report a few additional points for consideration.
> > ### Novelty
> > I thank the authors for clarifying their contributions from a technical prospective, and I confirm that I find the use of box embeddings valuable and well-founded, as I already mentioned in my review. Overall, I think that the contribution is clearer after your clarification. Although I only partially agree with the claim that this work is "applying box embeddings to Language Modeling for the first time" (see, for example, [1], [2], [3]), I acknowledge that the paper makes a step forward in extending box embeddings to the analysis of richer LLM prompts.
> >
> > ### Evaluation
> > - On the comparison with vector-based embeddings. I thank the authors for adding this comparison. I find difficult to asses whether a 16% is only a "modest" drop in similarity: would it be possible to compare vector- and box-embeddings also on prompts belonging to different "hierarchial clusters"? Is it possible to show how box-embeddings perform as we increase the distance between prompts, compared to vector-embeddings?
> > - On the comparison with hyperbolic and Poincaré embeddings. I thank the authors for adding references on this point. However, I still think that some comparison with i) hyperbolic methods or with ii) other state of the art techniques for specificity and inclusion is necessary and would make the claims significantly stronger.
> >
> >
> > [1] : Vilnis, Luke, et al. "Probabilistic embedding of knowledge graphs with box lattice measures."_Proceedings of the 56th Annual Meeting of the Association for Computational Linguistics (Volume 1: Long Papers)_. 2018.
> > [2] : Dasgupta, Shib, et al. "Word2box: Capturing set-theoretic semantics of words using box embeddings."_Proceedings of the 60th Annual Meeting of the Association for Computational Linguistics (Volume 1: Long Papers)_. 2022.
> > [3] : Li, Xiang, et al. "Smoothing the geometry of probabilistic box embeddings."_International conference on learning representations_. 2018.

---

> > > ### Author Response · Authors · 2026-04-08
> > >
> > > ## First time box for language modeling
> > > Thank you for pointing this out, we agree the original wording was slightly incorrect. To the best of our knowledge, this is the first work to successfully apply box embeddings to longer text. The prior works referenced in the review are already discussed in our paper; however, they primarily focus on words or entities in knowledge graphs. Existing work has never tried to capture sentence level semantics using boxes. Extending box embeddings to sentences is non-trivial, as it requires defining a meaningful interpretation of box volume and constructing training data that supports such semantics.
> > >
> > > ## Other entailment
> > > Regarding hyperbolic and Poincaré embeddings, applying them in this setting is not straightforward. Effectively leveraging their entailment properties would require carefully designed training objectives and data, which is beyond the scope of this work and goes away from the main point of the paper. However, in response to the suggestion, we include results using CSDelta, which models relationships via vector distances, and report its performance for comparison.
> > >
> > > | Model / Setting     | FollowBench (Acc) | STS-B (Spearman) | SURI (Acc) |
> > > |--------------------|-------------------|------------------|------------|
> > > | Vector             | 0.640             | **0.835**        | 0.725      |
> > > | Vector w/o entail  | 0.627             | 0.704            | 0.539      |
> > > | Box                | 0.738             | 0.760            | 0.868      |
> > > | Box w/o synth      | 0.716             | 0.756            | 0.889      |
> > > | Box w/o entail     | 0.687             | 0.727            | 0.640      |
> > > | Box w/o links      | **0.775**         | 0.661            | **0.924**  |
> > > |  CSDelta     | 0.012         | 0.757           | **0.95**  |
> > >
> > >
> > > ## Similarity
> > > Concerning the modest differences in similarity, we emphasize that we use an off-the-shelf similarity model with the same base encoder as the vector embeddings, which introduces an inherent bias. Additionally, the model is not fine-tuned specifically for prompt analysis. Nevertheless, following the reviewer’s suggestion, we extend our analysis using an alternative distance metric based on the depth of the earliest common ancestor. We also expand the evaluation to include all examples, rather than restricting to pairs with mutual neighbors. While this leads to a slight increase in the observed differences, the effect diminishes at larger distances. Finally, both visualizations are provided in the supplementary material at the location referenced in the paper.
> > >
> > > |Parent depth | Vector_sim | Box sim | Degradation |
> > > |---|---|---|---|
> > > | 2 | 0.296570 | 0.219534 | 25.9% |
> > > | 3 | 0.243734 | 0.182837 | 24.9% |
> > > | 4 | 0.210633 | 0.158778 | 24.6% |
> > > | 5 | 0.179586 | 0.146277 | 18.5% |
> > > | 6 | 0.160623 | 0.131307 | 18.25% |
> > > | 7 | 0.143906 | 0.125286 | 12.9% |
> > > | 8 | 0.129201 | 0.117392 | 9.1% |
> > > | 9 | 0.121073 | 0.109266 | 9.7% |

---

### Official Review · Reviewer_RXNT · 2026-03-12

**Soundness:** 2
**Presentation:** 2
**Significance:** 1
**Originality:** 2
**Overall Recommendation:** 2
**Confidence:** 3

**Summary:**

This paper proposes PROMPT2BOX, a framework for embedding LLM prompts into a box embedding space in order to capture both semantic similarity and specificity (entailment) relations simultaneously. The core motivation is well-grounded: existing LLM weakness analysis pipelines embed prompts as vectors and cluster them by topical similarity, but this collapses prompts that share a topic while differing in specificity and difficulty, making it hard to diagnose whether a model fails on a topic broadly or only on its more constrained variants.

**Compliance With Llm Reviewing Policy:**

Affirmed.

**Key Questions For Authors:**

- Box w/o links outperforms Box w/ links on every metric in Tables 1 and 2 except STS-B. What is the principled criterion for preferring the links model for downstream evaluation? Have the authors run the hierarchical clustering experiments in Section 7 using Box w/o links? If so, what are the results, and does it improve or worsen the weakness discovery and specificity-depth metrics?
- The specificity-depth metric relies on GPT-5.1-mini labels for which of two instructions is more specific. What is the estimated accuracy of these annotations? Have the authors spot-checked a sample of annotation pairs against human judgments? What fraction of the annotated pairs are unambiguous (e.g., one instruction is a strict lexical superset of the other) versus genuinely ambiguous (e.g., two different constraint dimensions), and does annotation quality differ between these cases?

**Limitations:**

yes

**Strengths And Weaknesses:**

Strengths
- The motivating problem is well-articulated. The observation that vector-based clustering conflates topically similar but difficulty-differentiated prompts is supported.
- A 70% vs. 52.71% specificity-depth agreement for box vs. vector hierarchical clustering is a meaningful gap. This validates the paper's core claim that boxes induce more meaningful hierarchies for the purpose of LLM weakness analysis, where depth should correspond to specificity.

Weaknesses
- The best-performing model (Box w/o links) is not used in the downstream evaluations, and this inconsistency is not adequately justified. Table 1 shows that Box w/o links achieves the best FollowBench (0.775) and SURI (0.787) scores. Table 2 shows it also achieves the best RMSE (1.4777). Yet Section 5.2 explicitly states "we will use the model trained with links dataset in the rest of the experiments." The justification given is that the model w/links is "most well-rounded" is informal. The paper should either use Box w/o links throughout and explain why the STS-B tradeoff is acceptable, or provide a principled criterion for model selection with quantitative support.
- Table 6 shows that box-based finds fewer size-2 weakness clusters than vector-based for 6 of 17 models. Table 7 shows that the box-based method finds fewer all-size weakness clusters than vector-based for 7 of 17 models, including starchat (−27.68%) and wizardlm-70b (−16.77%). The paper reports only the positive average without analyzing the failures. Why does box underperform vector-based approach for these specific models?
- Box embedding's advantage over vectors might shrink or disappear entirely because frontier models have more uniform performance across specificity levels. No evidence that this method can generalize to other scoring protocols, other prompt domains, or more capable models where the performance variance that the method exploits may not exist.

---

> ### Author Rebuttal · Authors · 2026-03-29
>
> We thank the reviewer for their response. We seek to answer major issues/complaints with the paper
>
> ## Choice of box w/links
>
> Our goal is for box embeddings to model both semantic relevance and specificity. However, without links, the box model suffers from significant degradation in semantic structure. We made this conclusion based on both qualitative visualizations and quantitative metrics such as STSB performance (a 13% drop). While the unlinked variant preserves entailment better then the linked variant, it substantially degrades semantic relevance, resulting in an worse modeling of the prompt space.
> We acknowledge that the choice between linked and unlinked variants appears somewhat arbitrary in the main text, and we will revise the presentation to clarify this design decision. We include new experiments with the “box w/o link” model in the hierarchical setting and will include them the appendix. The table is present in the rebuttal of LD6m. We observe that box w/o links performs worse across all settings, with the exception of local score consistency, where it shows a marginal gain of 0.8%.
>
> ## Why do vectors find more weaknesses sometimes?
>
> We believe this behavior primarily arises from the inherent stochasticity of LLMs. Even under the assumption of a “gold” hierarchical clustering algorithm, there is no guarantee that it will yield the minimum possible neighbor score differences or highest number of weakness clusters. A particular LLM might not give similar scores for prompts that are semantically or structurally similar leading to inconsistencies in local neighborhood relationships.
> As a result, there is an unavoidable degree of stochasticity in the evaluation process itself. To account for this, we conduct experiments across all 17 models in UltraFeedback. Our results show that, in the majority of cases, our approach outperforms baselines in all the metrics. We will clarify this in the main text.
>
> ## Correctness of LLM labels
> To validate the quality of LLM predictions, we first manually inspected a subset of outputs and found them to be largely reasonable. We additionally conduct a human evaluation study with four annotators.
>
> Each annotator is given the same set of 50 instruction pairs, randomly sampled from the full set of 864 examples. For each pair, annotators were asked: (1) which instruction is more specific, or whether they are equally specific, and (2) whether one instruction is a direct lexical superset. Since the sampled pairs are limited in size, we instructed annotators to mark containment as “yes” even in slightly relaxed cases where constraints are weakened but still largely preserved.
>
> We note that both specificity and containment judgments can be subjective. We find that the average inter-annotator agreement for specificity to be 45.3%. To account for this, we aggregate results by averaging across annotators. In cases of ties between “equally specific” and one instruction being more specific, we resolve in favor of one instruction (A or B), as the LLM does not produce an “equally specific” output. (Although we updated the LLM to allow this option, it was never selected by the LLM.)
>
> We observe that LLM agreement with average human judgments is approximately 59.2%, which is notably higher than the inter-annotator agreement of 45.3%. If we exclude cases where annotators selected “equally specific” (10 cases), LLM agreement increases to 74.3%.
>
> For containment, annotators judged that 64% of pairs exhibit either strict or slightly relaxed containment. LLM agreement is 61.2% for strict containment cases and 58.1% otherwise, indicating slightly better agreement in the case of containment.
>
> The results above indicate that LLM-generated annotations are reasonably reliable. We conduct an additional experiment where use these human annotations for the specificity depth agreement. Here the box embedding approach achieves an accuracy of 61.2%, significantly outperforming the vector embedding baseline at 32.7%.
>
> ## Box embedding's advantage over vectors
>
> Despite the strong performance of frontier models, recent work such as MOSAIC[1], CCR Bench[2], and CFBench[3] shows that these models continue to struggle as the number and complexity of constraints increase, with noticeable performance degradation on long and compositional instructions. Further, work, such as “LLMs Get Lost in Multi-Turn Conversation” [4], shows that multi-turn interactions become challenging for LLMs as constraints accumulate over turns.
>
> These trends directly motivate the use of box embeddings, which are inherently better suited to represent hierarchical structure and constraint inclusion.
>
> [1] Purpura et al. "MOSAIC" (https://aclanthology.org/2026.eacl-long.62/)
>
> [2] Xue et al. "CCR-Bench" (https://arxiv.org/abs/2603.07886)
>
> [3] Zhang and Zhu and Shen et al. "CFBench" (https://aclanthology.org/2025.acl-long.1581/)
>
> [4] Laban et al. "LLMs Get Lost in Multi-Turn Conversation" (https://arxiv.org/abs/2505.06120)

---

> > ### Author Rebuttal · Reviewer_RXNT · 2026-04-05
> >
> > Thank you for the rebuttal.
> > - W.1 (Box w/o links inconsistency): Partially resolved. The authors provide hierarchical clustering experiments with Box w/o links and justify the choice via STS-B degradation. However, the justification remains informal, and the rebuttal contains an inconsistency: the response to my review states Box w/o links "performs worse across all settings," while the results provided to Reviewer LD6m show it actually wins on local score consistency. This inconsistency undermines confidence in the model selection rationale.
> > - W.2 (Models where vectors outperform boxes): Not resolved. The explanation attributes underperformance to LLM stochasticity and lacks targeted analysis of why specific models such as starchat and wizardlm-70b show large negative gaps. This concern remains open.
> > - W.3 (Generalization to frontier models): Partially resolved. It is not directly demonstrated that box embeddings maintain their advantage on frontier models.
> > - Q.2 (Annotation quality): Partially resolved. The human evaluation is a welcome addition, but 45.3% inter-annotator agreement raises questions about the reliability of the specificity-depth metric as a ground truth signal.
> >
> > I will maintain my score.

---

> > > ### Author Response · Authors · 2026-04-08
> > >
> > > W.1: We gave the design choice, as well as detailed results backing up the claim. The "inconsistency", is talked about in the exact same line, where we talk about the fact that the difference is less than 1% better in exactly one situation.
> > >
> > > W3: We highlight recent work showing that even frontier models continue to underperform on highly specific instructions. This demonstrates the importance of capturing and modeling specificity. In this context, standard vector-based methods lack the representational capacity to model such structured specificity effectively, whereas box embeddings are explicitly designed to capture hierarchical and containment relationships, thus showing that they will still maintain their advantage for frontier models.
> > > Further, our results already include results from GPT 4 and GPT 3.5. However, direct demonstration with frontier models of the current generation is infeasible because datasets pairing frontier models with such prompts and responses are scarce.
> > >
> > > Q2: While we acknowledge that specificity judgments are inherently subjective, the observed patterns of agreement suggest that this subjectivity is limited and structured rather than arbitrary. Since individual annotators can make errors on a per-question basis, it is more informative to examine aggregate results. Our purpose in reporting the 45.3% inter-annotator agreement was to show that, despite individual subjectivity, agreement is substantially better than random(33%), and that LLM judgments align well with the aggregate.
> > > When analyzing the responses, 52% of cases show agreement among at least three of the 4 annotators. Additionally, 26% of cases (13/50) reflect evenly split (2–2) decisions between “equally specific” and “more/less specific,” which we attribute to minor interpretational differences—e.g., whether a constraint is perceived as slightly stronger or equivalent.
> > >
> > > Together, these categories account for 78% of all cases, indicating substantial consensus overall. Most of the remaining instances follow a 2–1–1 split, which is still relatively mild disagreement. Notably, the truly contradictory scenario, where two annotators judge one prompt as more specific and the other two judge it as less specific, occurs only once.
> > >
> > > Overall, this distribution demonstrates that specificity depth is a reasonably consistent and reliable signal, even under minimal annotation guidance.

---

### Official Review · Reviewer_LD6m · 2026-03-12

**Soundness:** 2
**Presentation:** 3
**Significance:** 2
**Originality:** 3
**Overall Recommendation:** 3
**Confidence:** 3

**Summary:**

This paper argues that vector-based prompt representations primarily capture topical similarity, but are less suited for modeling prompt specificity and entailment, which in turn limits fine-grained weakness analysis. The authors propose PROMPT2BOX, a box-embedding approach that represents prompt semantics via box centers and prompt generality/specificity via box extent, together with Box-SNE and a corresponding hierarchical clustering method. Experiments on UltraFeedback and related proxy evaluations suggest that the proposed representation outperforms vector baselines in capturing instruction specificity and identifying fine-grained LLM weaknesses.

**Compliance With Llm Reviewing Policy:**

Affirmed.

**Key Questions For Authors:**

1. How much of PROMPT2BOX’s advantage depends on large-scale synthetic supervision? Have the authors evaluated a reduced-supervision setting with less or no GPT-based synthesized data?
2. Why does Box w/o links outperform the full model on several metrics, while the full model appears more balanced on STS-B? Do the authors attribute this to a trade-off in objectives, the linkage data, or loss weighting?
3. Since Box-SNE enforces a 2D projection of high-dimensional box relations, how robust are the resulting visualizations to distortion? Are there cases where the 2D structure becomes misleading?

**Limitations:**

No. The paper does not include a clear dedicated discussion of limitations. It would be helpful to explicitly discuss the reliance on synthetic supervision, the use of proxy metrics for clustering evaluation, and possible distortion in Box-SNE visualizations.

**Strengths And Weaknesses:**

Strengths
1. The paper is well motivated: it clearly identifies that vector-based prompt representations mainly capture topical similarity, while box embeddings are better suited to modeling specificity and entailment.
2. The proposed method is fairly complete, covering representation learning, synthetic/existing training data, visualization with Box-SNE, and hierarchical clustering for weakness analysis.

Weaknesses
1. The method relies substantially on synthetic supervision, so the learned geometry may partly reflect the data construction pipeline rather than an independently validated prompt structure.
2. The results suggest a trade-off between modeling semantic similarity and modeling specificity/entailment, and the balance between the two objectives is not fully resolved.
3. The hierarchical clustering evaluation is based mostly on proxy metrics, since there is no direct evaluation against human-annotated prompt hierarchies.

---

> ### Author Rebuttal · Authors · 2026-03-29
>
> We thank the reviewer for their response. We seek to answer major issues/complaints with the paper
>
> ## Reliance on large scale synthetic datasets
>
> In the training process, the entailment data consists of both existing datasets and synthetically synthetic datasets. We use existing resources, SURI and MultiNLI to learn some of the entailment relationships. However these datasets do not provide supervision for prompt specificity. To bridge this gap, we construct hierarchical data from WildChat and link it with Infinity Instruct to resolve the drop in semantic relevance when training with the former.
>
> Based on the review, we conduct an additional ablation where the model is trained without any synthetic datasets. We then evaluate this model on the hierarchical clustering task to see the effect on prompt hierarchical structure and present the corresponding results below. We see that without the synthetic data, the model performs worse across practically all the metrics. We will add these results to the main paper.
>
>
> | Metric                          | Rand.  | Box w/o synth.   | Box      |
> |--------------------------------|--------|--------|----------|
> | Local Score Consist. Improv.   | 0.0%   | 10.21%  | **12.32%** |
> | LLM Spec-Depth Agreement    | 0.0%   | 66.03%  | **68.34%** |
> | Size 2 Weakness Improv.        | --     | 0.0%   | **1.41%**  |
> | All Size Weaknesses Improv.    | --     | 0.0%   | **2.31%**  |
>
> We also note that a some of the values for SURI in Table 1 were misreported in the original submission. We include a corrected Table 1 below, which also incorporates the results from box w/o synth.
>
> | Model / Setting     | FollowBench (Acc) | STS-B (Spearman) | SURI (Acc) |
> |--------------------|-------------------|------------------|------------|
> | Vector             | 0.640             | **0.835**        | 0.725      |
> | Vector w/o entail  | 0.627             | 0.704            | 0.539      |
> | Box                | 0.738             | 0.760            | 0.868      |
> | Box w/o synth      | 0.716             | 0.756            | 0.889      |
> | Box w/o entail     | 0.687             | 0.727            | 0.640      |
> | Box w/o links      | **0.775**         | 0.661            | **0.924**  |
>
>
>
> ## Box w/link clarification
>
> The reason for this is discussed in Section 3.4.5 of the paper. Box w/o links consists of hierarchical instructions synthesized from WildChat. While the addition of this data improved performance on entailment-related metrics, it led to a decline in the model's semantic understanding. As discussed in the main text, we hypothesize that there is a tradeoff between semantic understanding and the ability to model specificity, at least under the current training process. The hierarchical instructions bias the model toward prioritizing specificity over semantic relevance. We believe that this bias is the reason box w/o link has a lower RMSE than box w/link.
> We run the hierarchical clustering experiments with box_no_links and present the results below. We observe that box_no_links performs slightly worse across all settings, with the exception of local score consistency improvement, where it shows a marginal gain of 0.8%
>
> | Metric                          | Rand.  | Box w/o links.   | Box      |
> |--------------------------------|--------|--------|----------|
> | Local Score Consist. Improv.   | 0.0%   | **12.6%**  | 12.49% |
> | LLM Spec-Depth Agreement    | 0.0%   | 67.07%  | **68.34%** |
> | Size 2 Weakness Improv.        | --     | 0.0%   | **4.95%**  |
> | All Size Weaknesses Improv.    | --     | 0.0%   | **7.45%**  |
> | Local Neighbor Similarity Improv.   | --  | 0.0%  | **14.6%** |
>
>
> ## Box-SNE
>
> Distortion in low-dimensional projections is inevitable. This issue is further exacerbated for box embeddings compared to vector embeddings, as boxes are constrained not only in their position but also in the volume they can occupy, effectively reducing the available representational space.
>
> To the best of our knowledge, this is the first work to explore 2D projections of box embeddings. However, our approach is fundamentally based on SNE and therefore inherits its well-known limitations. We observe the common issue of overcrowding near the center of the projection, which is also evident in our visualizations.
>
> We will explicitly acknowledge these limitations in the revised manuscript. We also hope that this work motivates future research toward improved methods for visualizing box embedding spaces with reduced distortion.
>
> ## Use of Proxy Metrics
>
> Gathering human annotations for hierarchical clustering of boxes is impractical due to the high variance and subjectivity involved in such evaluations. We therefore rely on a set of proxy metrics grounded in desirable properties of hierarchical clustering, such as lower neighbor score differences, improved weakness capturing and higher depth-specificity correlation. We will acknowledge the reliance on proxy metrics in the limitations section of the paper.

---

> > ### Author Rebuttal · Reviewer_LD6m · 2026-04-04
> >
> > Thank you for the detailed rebuttal and the additional ablations. The responses are helpful and address my questions to some extent, but my main concerns are only partially resolved, particularly regarding the reliance on synthetic supervision and the use of proxy metrics. Therefore, I will keep my current score unchanged.

---

> > > ### Author Response · Authors · 2026-04-08
> > >
> > > We thank the reviewer for their response. While analyzing the quality and diversity of the synthetic data is indeed important, we would like to clarify that it is not the primary contribution of this work. Also we have shown in the rebuttal that the model trained without synthetic data only slightly trails behind in the benchmarks and can still effectively capture specificity.
> > >
> > > The role of the synthetic dataset is to enable the model to learn hierarchical structure in prompts, a property that is largely absent in existing datasets. Finally, we emphasize that the synthetic data is constructed exclusively from WildChat, which is not included in any of the evaluation benchmarks, thereby avoiding any risk of data contamination.

---

### Official Review · Reviewer_EQif · 2026-03-13

**Soundness:** 3
**Presentation:** 3
**Significance:** 3
**Originality:** 3
**Overall Recommendation:** 4
**Confidence:** 3

**Summary:**

This paper proposes Prompt2Box, a method for embedding prompts into a box embedding space in order to capture hierarchical entailment relationships between prompts. Existing analyses of prompt datasets typically rely on vector embeddings and clustering techniques, which primarily capture topical similarity but fail to represent relationships such as specificity or entailment. As a result, prompts that share a similar topic but differ in specificity are often treated as similar in embedding space, limiting the ability to analyze model weaknesses. To address this limitation, the authors introduce a prompt encoder that maps prompts into box embeddings, where each prompt is represented as a hyper-rectangle in latent space. This representation allows inclusion relationships between boxes to capture entailment structures, such as more specific prompts being nested within more general ones. The model is trained on both existing and synthetically generated prompt pairs to learn these hierarchical relationships. The authors further introduce a dimension reduction method designed specifically for box embeddings to enable visualization and dataset analysis. Experiments on prompts derived from the UltraFeedback dataset show that Prompt2Box better captures prompt specificity relationships compared to vector embedding baselines. The method also improves the ability to analyze hierarchical weaknesses across multiple LLMs.

**Compliance With Llm Reviewing Policy:**

Affirmed.

**Key Questions For Authors:**

How sensitive is the learned prompt hierarchy to the choice of synthetic training data used during model training?
Can the discovered prompt hierarchies be used to improve dataset construction or evaluation benchmarks for LLMs?
How does Prompt2Box scale to very large prompt datasets compared to standard embedding methods?

**Limitations:**

YES

**Strengths And Weaknesses:**

[Strengths]

Soundness
The paper identifies a clear limitation of existing prompt analysis methods: vector embeddings capture semantic similarity but often fail to represent hierarchical relationships such as specificity. The proposed box embedding framework provides a natural way to model such relationships by representing prompts as regions rather than points in latent space.
The training approach, which uses both existing prompt data and synthesized examples to learn entailment relationships, is reasonable and aligns with prior work on box embeddings. The experiments show that Prompt2Box can better capture hierarchical prompt relationships compared to standard embedding approaches.

Presentation
The paper is clearly written and the motivation is easy to follow. The examples illustrating prompt specificity relationships help clarify why vector embeddings are insufficient for this task. The figures demonstrating the hierarchical relationships captured by box embeddings also improve the readability of the method.
The proposed dimensionality reduction technique for box embeddings is described clearly and allows intuitive visualization of prompt hierarchies.

Significance
Understanding the weaknesses of LLMs through prompt analysis is an important research direction. Tools that reveal structural relationships between prompts can help researchers identify systematic model failures and better design evaluation datasets. By modeling prompt entailment structures explicitly, this work contributes a useful tool for analyzing prompt datasets and evaluating LLM behavior.

Originality
The main novelty lies in applying box embeddings to model entailment relationships among prompts. While box embeddings have been used in other contexts such as knowledge representation and hierarchical reasoning, their application to prompt analysis and LLM evaluation is a creative extension.

[Weaknesses]

Limited downstream impact
Although the method improves the ability to represent prompt specificity relationships, the practical implications for improving LLM performance remain somewhat indirect. The paper primarily focuses on analysis rather than demonstrating how the discovered structures can lead to better models or training strategies.

Dependence on synthetic training data
The training process relies partly on synthesized prompt pairs to learn entailment relationships. The quality and diversity of these synthetic examples may influence the learned structure, but the paper provides limited analysis of how this affects the results.

Evaluation scope
While the experiments demonstrate improved hierarchical structure discovery, the evaluation is mostly limited to prompt clustering and visualization tasks. Additional downstream applications, such as improved dataset construction or evaluation benchmarks, would strengthen the empirical impact.

---

> ### Author Rebuttal · Authors · 2026-03-29
>
> We thank the reviewer for their response. We seek to answer major issues/complaints with the paper
>
> ## Reliance on synthetic datasets
>
> Learning prompt hierarchy and structure is an inherently challenging problem, and to the best of our knowledge, there is no prior work that directly addresses this task. Consequently, there is a lack of suitable datasets that capture hierarchical relationships between prompts.
>
> While we leverage existing resources such as SURI and MultiNLI to learn entailment relationships, these datasets do not provide supervision for prompt specificity. To bridge this gap, we construct hierarchical data from WildChat and link it with Infinity Instruct to resolve the drop in semantic relevance when training with the former.
>
> Although synthetic data is used during training, our evaluation is conducted on separate datasets: FollowBench and UltraFeedback. This ensures that performance gains are not limited to synthetic distributions.
>
> To measure the effect of synthetic data on the prompt structure, we conduct an additional ablation where the model is trained without any synthetic datasets. We then evaluate this model on the hierarchical clustering task and present the corresponding results below. We see that without the synthetic data, the model performs worse across practically all the metrics. The number for LLM Spec-Depth Agreement is different as the initial algorithm used hard intersection instead of soft intersection which the model was trained for, leading to slight changes in the values. Further we fixed a bug with the code which caused some of the responses to not return a value. We also report the new Table 2 with this.
>
>
> | Metric                          | Rand.  | Box no synth.   | Box      |
> |--------------------------------|--------|--------|----------|
> | Local Score Consist. Improv.   | 0.0%   | 10.21%  | **12.32%** |
> | LLM Spec-Depth Agreement    | 0.0%   | 66.03%  | **68.34%** |
> | Size 2 Weakness Improv.        | --     | 0.0%   | **1.41%**  |
> | All Size Weaknesses Improv.    | --     | 0.0%   | **2.31%**  |
>
>
> | Metric                          | Rand.  | Vector   | Box      |
> |--------------------------------|--------|--------|----------|
> | Local Score Consist. Improv.   | 0.0%   | 9.25%  | **12.82%** |
> | LLM Spec-Depth Agreement    | 0.0%   | 51.76%  | **68.34%** |
> | Size 2 Weakness Improv.        | --     | 0.0%   | **7.70%**  |
> | All Size Weaknesses Improv.    | --     | 0.0%   | **13.47%**  |
>
> We also note that several values for SURI in Table 1 were misreported in the original submission. We include a corrected Table 1 below, which also incorporates the results from box w/o synth.
>
> | Model / Setting     | FollowBench (Acc) | STS-B (Spearman) | SURI (Acc) |
> |--------------------|-------------------|------------------|------------|
> | Vector             | 0.640             | **0.835**        | 0.725      |
> | Vector w/o entail  | 0.627             | 0.704            | 0.539      |
> | Box                | 0.738             | 0.760            | 0.868      |
> | Box w/o synth      | 0.716             | 0.756            | 0.889      |
> | Box w/o entail     | 0.687             | 0.727            | 0.640      |
> | Box w/o links      | **0.775**         | 0.661            | **0.924**  |
>
>
> ## Performance on downstream tasks.
> Through experiments on hierarchical clustering and RMSE on UltraFeedback, we demonstrate that our learned representation space is better suited for evaluating LLM performance, capturing both semantic similarity and fine-grained differences in prompt specificity.
>
> The main purpose of the paper is to show the importance of modeling specificity for prompt understanding. While constructing downstream applications is beyond the scope of this paper, we note that prior works such as SkillVerse[1] and EvalTree[2] explicitly rely on hierarchical clustering to identify model weaknesses and perform weakness guided data collection. In this context, improvements in the quality of hierarchical structure directly translate to stronger downstream evaluation pipelines, and we show via multiple proxy metrics that our method produces more meaningful hierarchical clusters compared to baselines.
>
> [1] Tian et al. "SkillVerse : Assessing and Enhancing LLMs with Tree Evaluation"
>
> [2] Zeng et al. "EVALTREE: Profiling Language Model Weaknesses via Hierarchical Capability Trees"

---

> > ### Author Rebuttal · Reviewer_EQif · 2026-04-04
> >
> > The rebuttal provides useful clarification on the role of synthetic training data and includes additional analysis, which helps address part of my concern.
> > However, my original concern regarding the sensitivity to the quality and diversity of synthetic data is not fully resolved and would likely require more substantial analysis beyond the scope of a short rebuttal. That said, I still find the problem well-motivated and the overall contribution meaningful, and therefore I maintain my original score.

---

> > > ### Author Response · Authors · 2026-04-08
> > >
> > > We thank the reviewer for their response. While analyzing the quality and diversity of the synthetic data is indeed important, we would like to clarify that it is not the primary contribution of this work. Also we have shown in the rebuttal that the model trained without synthetic data only slightly trails behind in the benchmarks and can still effectively capture specificity.
> > >
> > > The role of the synthetic dataset is to enable the model to learn hierarchical structure in prompts, a property that is largely absent in existing datasets. Finally, we emphasize that the synthetic data is constructed exclusively from WildChat, which is not included in any of the evaluation benchmarks, thereby avoiding any risk of data contamination.

---

### Decision · Program_Chairs · 2026-04-30

**Decision:**

Reject

**Comment:**

This paper studies how to represent LLM prompts in a way that captures not only semantic similarity but also entailment and specificity relations. The proposed method embeds prompts as boxes rather than points, and uses this representation for hierarchical clustering and weakness analysis of LLMs.

Reviewers generally agreed that the problem is well motivated, that the use of box embeddings is a natural fit for modeling asymmetric prompt relations, and that the empirical results show improvements over baselines

However, one major concern is that the paper does not yet sufficiently validate the proposed method beyond the setting most favorable to it. Reviewers EQif and LD6m both raised concerns about the reliance on synthetic supervision, and both remained unconvinced that the rebuttal fully resolved questions about how much the learned geometry depends on the synthetic data construction pipeline. Reviewer LD6m also highlighted that the hierarchical clustering results are evaluated mostly through proxy metrics rather than direct human-validated hierarchies.

The second major concern is evaluation and model selection. Reviewer RXNT questioned the rationale for using the linked box model in downstream tasks when the unlinked variant performs better on several upstream metrics, and found the rebuttal only partially resolved this. He/She also noted that the paper does not adequately analyze cases where vector methods outperform the box approach for specific models.

One remaining concern is the novelty. Reviewer yYvq acknowledged that the contribution became clearer after the rebuttal, but still questioned the degree of technical novelty beyond applying box embeddings to a new problem setting, and also noted the absence of stronger comparisons to alternative structured embedding methods. These concerns were only partially resolved.

Overall, while the paper presents an interesting and potentially useful perspective on learning the prompt structure, the current submission would benefit from stronger validation of the learned hierarchy and broader comparison to contenders before it is ready for publication at ICML.